# Persons tested for SAR-CoV-2 at a military treatment facility in Hawaii

**Javier Barranco-Trabi**[1]*, **Stephen Morgan**[2], **Seema Singh**[2], **Jimmy Hill**[2],
**Alexander Kayatani**[3], **Victoria Mank**[1], **Holly Nesmith**[4], **Heather Omara**[4], **Louis Tripoli**[1],
**Michael Lustik**[5], **Jennifer Masel**[6], **Sharon Chi**[6], **Viseth Ngauy**[6]

1 Department of Internal Medicine, Tripler Army Medical Center (TAMC), Honolulu, Hawaii, United States of America, 2 Department of Infectious Control, Tripler Army Medical Center (TAMC), Honolulu, Hawaii, United States of America, 3 Department of Microbiology, Tripler Army Medical Center (TAMC), Honolulu, Hawaii, United States of America, 4 Department of Family Medicine, Tripler Army Medical Center (TAMC), Honolulu, Hawaii, United States of America, 5 Department of Clinical Investigation, Tripler Army Medical Center (TAMC), Honolulu, Hawaii, United States of America, 6 Department of Infectious Disease, Tripler Army Medical Center (TAMC), Honolulu, Hawaii, United States of America

* javier.j.barranco@gmail.com

**Data Availability Statement:** All relevant data are within the paper and its Supporting information files.

**Funding:** The author(s) received no specific funding for this work.

## Abstract

Health inequalities based on race are well-documented, and the COVID-19 pandemic is no exception. Despite the advances in modern medicine, access to health care remains a primary determinant of health outcomes, especially for communities of color. African-Americans and other minorities are disproportionately at risk for infection with COVID-19, but this problem extends beyond access alone. This study sought to identify trends in race-based disparities in COVID-19 in the setting of universal access to care. Tripler Army Medical Center (TAMC) is a Department of Defense Military Treatment Facility (DoD-MTF) that provides full access to healthcare to active duty military members, beneficiaries, and veterans. We evaluated the characteristics of individuals diagnosed with SARS-CoV-2 infection at TAMC in a retrospective, case-controlled (1:1) study. Most patients (69%) had received a COVID-19 test within 3 days of symptom onset. Multivariable logistic regression analyses were used to identify factors associated with testing positive and to estimate adjusted odds ratios. African-American patients and patients who identified as "Other" ethnicities were two times more likely to test positive for SARS-CoV-2 relative to Caucasian patients. Other factors associated with testing positive include: younger age, male gender, previous positive test, presenting with >3 symptoms, close contact with a COVID-19 positive patient, and being a member of the US Navy. African-Americans and patients who identify as "Other" ethnicities had disproportionately higher rates of positivity of COVID-19. Although other factors contribute to increased test positivity across all patient populations, access to care does not appear to itself explain this discrepancy with COVID-19.

## Background

First identified in December 2019 in Wuhan, China, SARS-COV2 quickly spread to the rest of the world [1]. COVID-19 disease has been shown to disproportionately affect communities of color, including Asian Americans and Pacific Islanders, with resultant impact on clinical

**Competing interests:** The authors have declared that no competing interests exist.

outcomes attributable to limited access to care and the presence of co-morbid conditions [2–4]. Situated in the middle of the Pacific, Hawaii is of strategic importance to the Department of Defense (DOD) with over 300,000 DOD ethnically diverse beneficiaries living and working in the region, many of whom receive their medical care through Tripler Army Medical Center (TAMC). As the pandemic raged, access to health care was of concern, especially for many communities of color. Hawaii is unique in that there is a large Asian American and Pacific Islander population. Within the DOD, regardless of race/gender, medical coverage is provided through a universal health care plan (Tricare) for all service members, their families, and retirees.

The state of Hawaii and regional DOD officials implemented strict public health and infection control practices such as stay at home orders, mandatory masking, social distancing, and travel restrictions off/on the island with mandatory quarantine. TAMC implemented entry checkpoints, visitor restrictions, and liberal teleworking policies to minimize nosocomial spread. Virtual telehealth medical visits were implemented for most outpatient encounters to limit the spread of COVID-19 in the hospital. TAMC obtained and implemented numerous emergency use authorization (EUA) diagnostic platforms early on (e.g. ABI-7500, Biofire, Panther/Panther Fusion, Cepheid) for the detection of SARS-COV2 in symptomatic and asymptomatic individuals for clinical testing, Force Health Protection testing, and pre-operative/pre-admission testing. Many patients received their COVID19 testing at drive-through or field locations without an associated provider visit [4, 5].

In the early phase of the pandemic, little was known about the clinical presentation, asymptomatic infection, transmission, and real-world accuracy of EUA diagnostics. The robust testing capabilities provided a unique opportunity to evaluate these variables in a highly diverse population with reliable access to healthcare. With our study, we aimed to evaluate the epidemiology, clinical manifestation, and demographic characteristics of individuals diagnosed with SARS-COV2 infection at TAMC, assess risk factors contributing to transmission and disease progression, and assess the real-world performance of SARS-COV2 diagnostic platforms approved under Emergency Use Authorization (EUA).

## Methods

Before the study began, it was determined to be exempt from IRB review by the Deputy Institutional Officer. This study is a review retrospective, case-controlled (1:1) study. A case was defined as having a positive test via one of the polymerase chain reaction (PCR) testing platforms, while a control was defined as having a negative test via the same platform. We conducted a review of the electronic medical records of patients seen at TAMC between 1 March 2020 to 15 September 2020 to identify variables such as age, gender, BMI, clinical symptoms, travel history, exposure risk, time to testing, and indication for testing. All data were fully anonymized before researcher accessed them to ensure patient's confidentiality. Cycle Threshold (Ct) values were obtained from our CAP-certified clinical lab; these values were not released to clinicians and were not used to guide clinical decisions. Multivariable logistic regression analyses were used to identify factors associated with testing positive and to estimate adjusted odds ratios.

## Results

We found that most (69%) patients received a COVID-19 test within 3 days of symptom onset (Fig 1). Patients who tested negative were twice as likely to have zero symptoms as those who tested positive (56% v. 26%, p<0.001), and patients who tested positive were almost twice as likely to present with $\geq$ 3 symptoms (Fig 2) than those who tested negative (46% v. 24%,

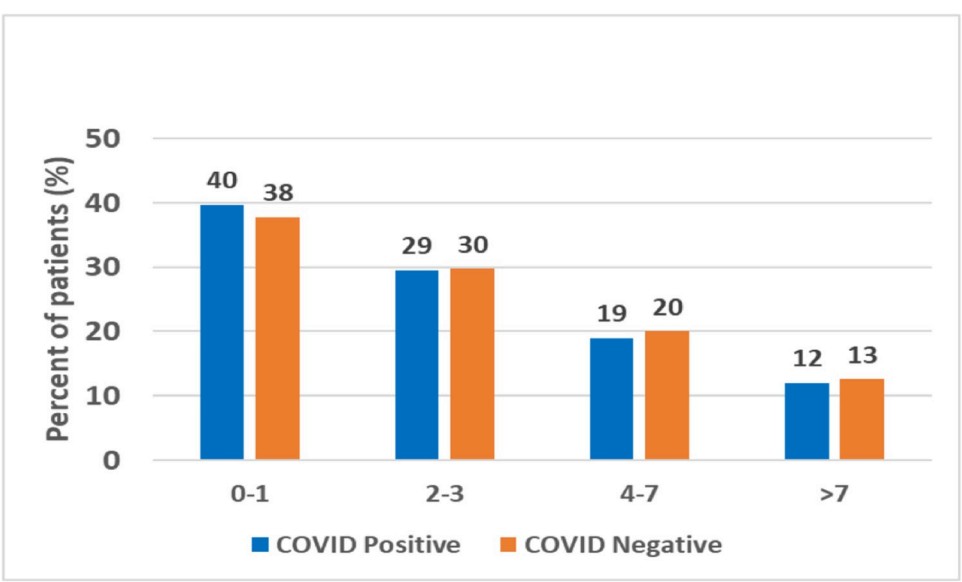

**Fig 1. Days between onset of symptoms and specimen collection date.**

p<0.001). Symptomatic patients with close contact to COVID-19 positive patients were 4 times more likely to test positive than those with symptoms but no close contacts (OR = 3.76) (Figs 3 and 4). As expected, lower rates of positivity were seen in asymptomatic individuals tested for screening purposes. Interestingly, nearly half these individuals tested for screening were positive on these highly sensitive tests.

African-American patients were two times more likely to test positive for SARS-COV2 relative to Caucasians patients; however, no difference in positivity was seen in Asian-American/Pacific Islanders relative to Caucasians patients (Fig 4). Cough, myalgia, fever, and anosmia were most associated with COVID-19 infection. In our test pool, active duty military made up

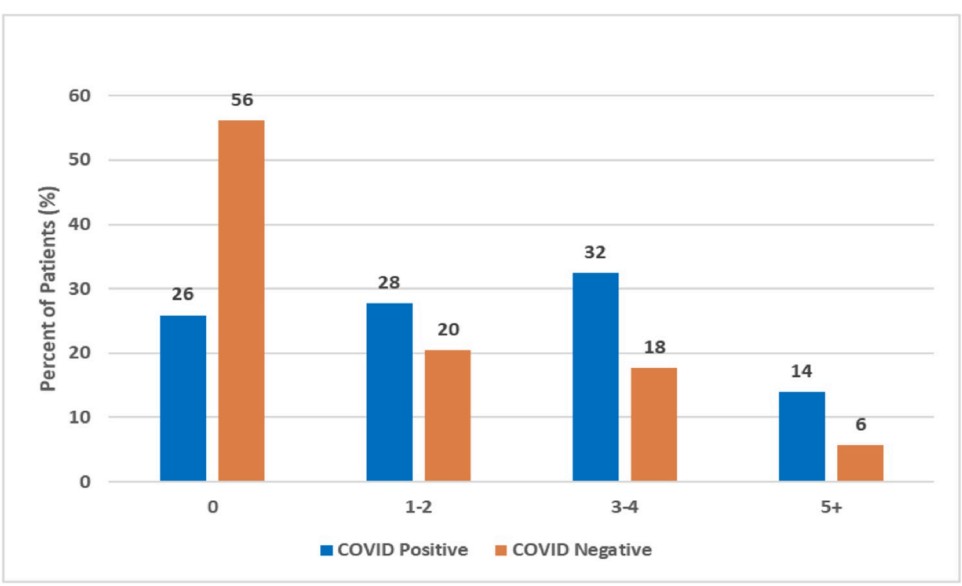

**Fig 2. Number of symptoms with Covid positive and negative test.**

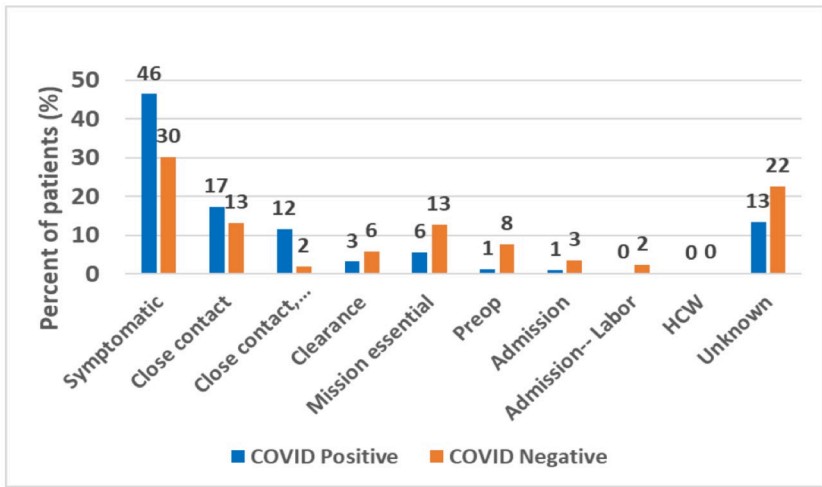

**Fig 3. Reason for testing.**

63% of all patients tested. Among with an identified race, Caucasians represented 62% of all tests, 17% African-American, 9% Asian American/Pacific Islanders, and 11% other. The percentage tested is representative of the Tricare beneficiary enrollment demographic at TAMC. We found that US Navy personnel were most likely to test positive (OR 1.42) and US Marines were least likely to test positive (OR 0.47) relative to US Army personnel.

Civilian personnel comprised the highest positivity rate relative to US Army personnel (OR 4.32 Fig 5). Among active duty personnel, male gender and African-Americans/other race had higher positivity rates. We also found that patients with higher BMI were less likely to test positive than those with lower BMI (Figs 4 and 6).

May-June 2020 saw the lowest positivity rates since March 2020 with a subsequent rise Aug/Sep 2020 timeframe, correlating to tightening and loosening of COVID-19 restrictions. Lower cycle threshold values were seen soon after symptom onset with a slow decline over two weeks (Fig 7).

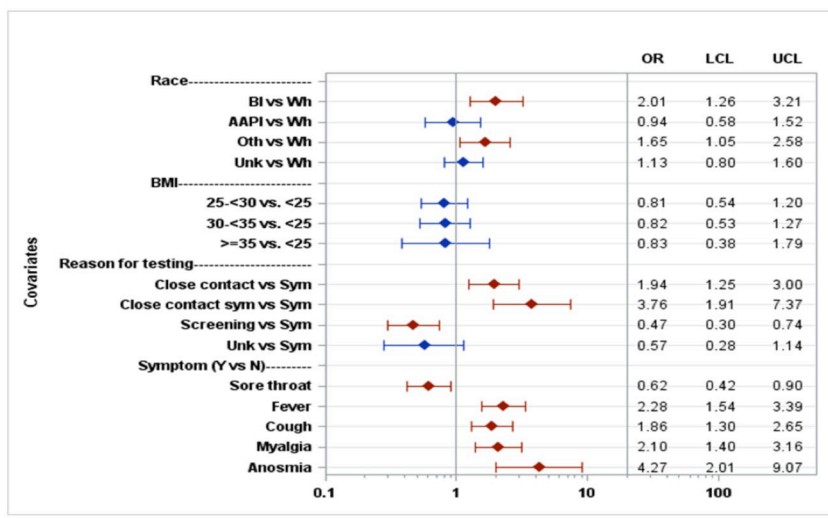

**Fig 4. Adjusted ORs for positivity based on case-control sample.**

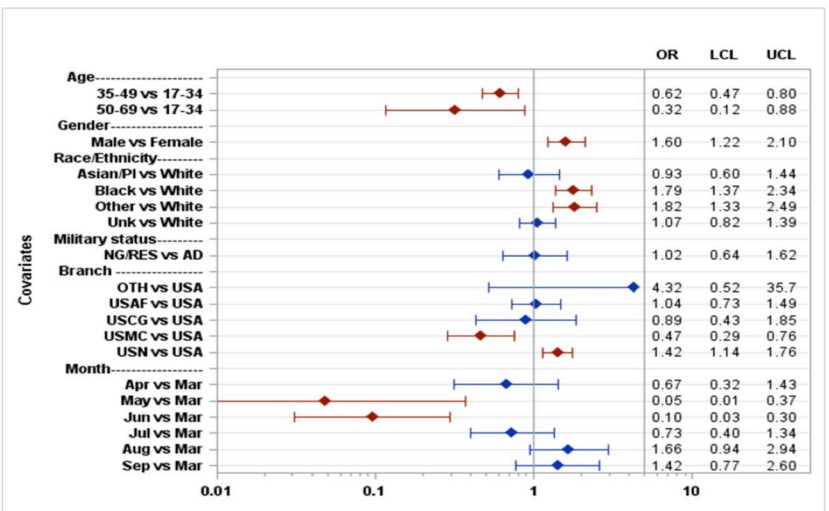

**Fig 5. Adjusted ORs for positivity based on all active duty service member test case-control sample.**

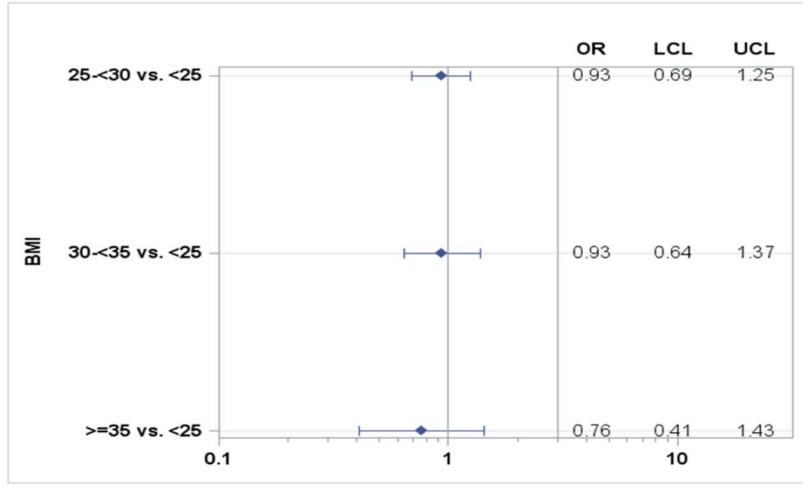

**Fig 6. Unadjusted BMI ORs for positivity based on case-control sample.**

## Discussion

Access to healthcare for rapid and accurate diagnosis of COVID-19 infection is one of the cornerstones of infection control and prevention. Cases that are identified rapidly can be appropriately instructed to quarantine to prevent the spread of disease. This facility has a unique benefit of a diverse population of US military, their beneficiaries, and retirees who have universal access to healthcare. We recognize that access to care remains a primary determinant of health outcomes, and this is particularly true for communities of color. We were able to compare rates of COVID-19 positivity in a diverse population with universal access to healthcare, identifying that African-Americans were two times more likely to test positive than their Caucasian counterparts despite no differences in access to care. Further study is warranted to determine the underlying mechanisms at play behind this racial disparity in COVID-19 rates.

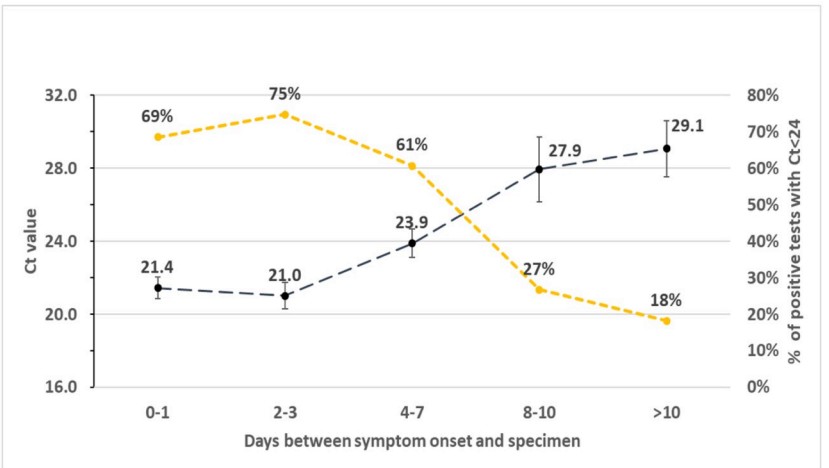

**Fig 7. Mean CT and % CT <24 by days between symptom onset and specimen.**

Our study also confirmed data previously published that patients who tested negative were twice as likely to be asymptomatic [6, 7]. In addition, we found that patients with close contact with a COVID-19 positive individual were more likely to test positive. This finding is not surprising given what we know about viral transmission.

Lastly, our study was able to compare COVID-19 positivity rates in patients with obesity. Obesity was defined as BMI >30. Much like COVID-19, obesity is a global disease. Previous studies have found that obese patients are at higher risk of mortality from COVID-19 [8]. In this study, we found that patients with obesity had lower COVID-19 positivity rates. Further research is needed in order to understand these results.

## Conclusion

Clinical symptoms of fever, cough, myalgia, and anosmia were most associated with COVID-19 infection in our patient population; this is consistent with the literature. Overall positivity rate of all person tested was 3.2%. African-Americans had disproportionately higher rates of positivity even in the setting of universal access to medical care. Obesity was not associated with higher positivity rate. However, there was a slightly lesser chance of having a positive test with a BMI of > 35. Routine screening in asymptomatic individuals, as expected, had lower positivity rates, suggesting infrequent asymptomatic infection/transmission. We did not identify any nosocomial transmission in our hospital in staff or patients. This is likely a result of the strict enforcement of infection control policies and enhanced testing. Universal access to care and robust diagnostic capabilities at TAMC contributed to the short interval between development of symptoms and COVID-19 testing and contributed to low infection rates in DOD beneficiaries. The use of telehealth visits, implementation of COVID-19 restrictions, and stay-at -home mandates limited provider visits so minimal clinical information was available in the medical records to help us refine our analysis of certain variables such as BMI and risk factors for infection.

## Supporting information

**S1 Table. Characteristics of COVID positive and COVID negative patients.**
(XLSX)

**S2 Table. Symptoms prevalence data.**
(XLSX)

**S1 Graph. OR BMI.**
(XLSX)

**S2 Graph. BMI.**
(XLSX)

## Author Contributions

**Conceptualization:** Javier Barranco-Trabi, Stephen Morgan, Seema Singh, Victoria Mank, Holly Nesmith, Heather Omara, Louis Tripoli, Viseth Ngauy.

**Data curation:** Javier Barranco-Trabi, Stephen Morgan, Jimmy Hill, Alexander Kayatani, Victoria Mank, Holly Nesmith, Heather Omara, Louis Tripoli, Michael Lustik, Jennifer Masel, Sharon Chi, Viseth Ngauy.

**Formal analysis:** Javier Barranco-Trabi, Seema Singh, Jimmy Hill, Alexander Kayatani, Victoria Mank, Holly Nesmith, Heather Omara, Louis Tripoli, Michael Lustik, Jennifer Masel, Sharon Chi, Viseth Ngauy.

**Funding acquisition:** Louis Tripoli.

**Investigation:** Javier Barranco-Trabi, Stephen Morgan, Jimmy Hill, Alexander Kayatani, Victoria Mank, Holly Nesmith, Heather Omara, Louis Tripoli, Michael Lustik, Jennifer Masel, Sharon Chi, Viseth Ngauy.

**Methodology:** Javier Barranco-Trabi, Stephen Morgan, Seema Singh, Jimmy Hill, Alexander Kayatani, Holly Nesmith, Heather Omara, Michael Lustik, Jennifer Masel, Sharon Chi, Viseth Ngauy.

**Project administration:** Javier Barranco-Trabi, Stephen Morgan, Seema Singh.

**Resources:** Victoria Mank.

**Supervision:** Javier Barranco-Trabi, Stephen Morgan, Seema Singh, Alexander Kayatani, Jennifer Masel, Viseth Ngauy.

**Validation:** Jennifer Masel, Sharon Chi, Viseth Ngauy.

**Visualization:** Jimmy Hill, Viseth Ngauy.

**Writing – original draft:** Javier Barranco-Trabi, Stephen Morgan, Seema Singh, Viseth Ngauy.

**Writing – review & editing:** Javier Barranco-Trabi, Stephen Morgan, Seema Singh, Viseth Ngauy.

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
