## [Decision Letter · Decision Letter 0]

26 Nov 2021

PONE-D-21-35130PERSONS TESTED FOR SAR-CoV-2 AT A MILITARY TREATMENT FACILITY IN HAWAIIPLOS ONE

Dear Dr. Barranco-Trabi,

Thank you for submitting your manuscript to PLOS ONE. After careful consideration, we feel that it has merit but does not fully meet PLOS ONE’s publication criteria as it currently stands. Therefore, we invite you to submit a revised version of the manuscript that addresses the points raised during the review process.

We look forward to receiving your revised manuscript.

Kind regards,

Sanjay Kumar Singh Patel, Ph.D.

Academic Editor

PLOS ONE

Journal Requirements:

2. Please ensure that you refer to Figure 7 in your text as, if accepted, production will need this reference to link the reader to the figure.

Reviewers' comments:

Reviewer's Responses to Questions

**Comments to the Author**

1. Is the manuscript technically sound, and do the data support the conclusions?

Reviewer #1: Yes

Reviewer #2: Yes

2. Has the statistical analysis been performed appropriately and rigorously? 

Reviewer #1: Yes

Reviewer #2: Yes

3. Have the authors made all data underlying the findings in their manuscript fully available?

Reviewer #1: Yes

Reviewer #2: Yes

4. Is the manuscript presented in an intelligible fashion and written in standard English?

Reviewer #1: Yes

Reviewer #2: Yes

5. Review Comments to the Author

Reviewer #1: In the current research article entitled " Persons Tested for Sar-CoV-2 at a Military Treatment Facility in Hawaii", by Barranco-Trabiet al., studied to identify trends in race-based disparities in COVID-19 in the setting of universal access to care. They evaluated the characteristics of individuals diagnosed with SARS-CoV-2 infection at TAMC in a retrospective, case-controlled (1:1) study by using Multivariable logistic regression analyses. Authors concluded that the African-American patients and patients who identified as “Other “ethnicities were two times more likely to test positive for SARS-CoV-2 relative to Caucasian patients. This article addresses a research topic of great interest, which is under intense investigation in the past years and the manuscript is generally well-written. However, this reviewer has certain suggestions that would help produce a more comprehensive overview of the topic:

Suggestions: -

1, Figures quality may be improved with high resolution images (minor).

2, The English of manuscript can be refined (minor).

3, It is required to provide an additional illustrative figure as to highlight the summary or prospect of this study.

4, More discussion needed to explain the findings.

5, Authors should cite some research to strengthen their hypothesis.

Reviewer #2: IIn this paper entitled "Persons tested for SAR-CoA-2 at a military treatment facility in Hawaii", the authors investigate a trend in race-based disparities in COVID-19 in individuals diagnosed with SARS-COV-2 infection in a retrospective, case-controlled study. The results are exciting and easy to understand. Many studies have found similar results in different settings. The manuscript is acceptable for publication. However, there is notable problem with the manuscript.

Comments:

1) The bar graphs in the manuscript (figure1,2,& 3) are regular Microsoft excel graphs. They should be changed for publication.

2) Discussion is completely missing from the manuscript. Please discuss our results.

3) Figure legends are also missing from the manuscript.

4) Proper reference is not provided in the manuscript.

---

## [Author Response · Author response to Decision Letter 0]

10 Jan 2022

Dear Editors, 

Thank you for inviting us to submit a revised draft of our manuscript entitled, " Persons tested for SAR-CoV-2 at a military treatment facility in Hawaii" to PLOS ONE. We also appreciate the time and effort you and each of the reviewers have dedicated to providing insightful feedback on ways to strengthen our paper. Thus, it is with great pleasure that we resubmit our article for further consideration. We have incorporated changes that reflect the detailed suggestions you have graciously provided. We also hope that our edits and the responses we provide below satisfactorily address all the issues and concerns you and the reviewers have noted.

To facilitate your review of our revisions, the following is a point-by-point response to the questions and comments delivered in your letter dated Dec 9th, 2021.

EDITOR SUGGESTIONS:

1. Please ensure that your manuscript meets PLOS ONE's style requirements

• RESPONSE: The editing changes were made based on PLOS ONE’s style requirements.

2. Please ensure that you refer to Figure 7 in your text as, if accepted, production will need this reference to link the reader to the figure.

• RESPONSE: Thank you for bring this up. Figure 7 has been added to the text. 

REVIEWER 1 COMMENTS:

4. Figures quality may be improved with high resolution images (minor).

• RESPONSE: We increased the image quality of our graphs. 

5. The English of manuscript can be refined (minor).

• RESPONSE: We review the manuscript and refined the English. Grammar changes has been made. Thank you for your feedback. 

6. It is required to provide an additional illustrative figure as to highlight the summary or prospect of this study.

• RESPONSE: Thank you for this suggestion; however, we already provide 7 graphs with our results. 

7. More discussion needed to explain the findings.

• RESPONSE: We included a discussion section. 

8. Authors should cite some research to strengthen their hypothesis

• Response: Thank you for your suggestion, this has been addressed. We included several studies to support our findings. 

REVIEWER 2 COMMENTS:

9. The bar graphs in the manuscript (figure1,2,& 3) are regular Microsoft excel graphs. They should be changed for publication.

• RESPONSE: All figures are Microsoft excel graphs. We are open to suggestions to programs that are free. Meanwhile, we increase the quality of our image to better represents our graphs. 

10. Discussion is completely missing from the manuscript. Please discuss our results

• RESPONSE: Discussion section was added. 

11. Figure legends are also missing from the manuscript.

• RESPONSE: Legends were added

12. Proper reference is not provided in the manuscript.

• RESPONSE: Reference were added to the manuscript

Again, thank you for giving us the opportunity to strengthen our manuscript with your valuable comments and queries. We have worked hard to incorporate your feedback and hope that these revisions persuade you to accept our submission.

Please address all correspondence concerning this manuscript to me at Javier.j.barranco@gmail.com

Thank you for your consideration of this manuscript.

Javier Barranco-Trabi, MD

Tripler Army Medical Center

Internal Medicine

---

## [Decision Letter · Decision Letter 1]

20 Jan 2022

PERSONS TESTED FOR SAR-CoV-2 AT A MILITARY TREATMENT FACILITY IN HAWAII

PONE-D-21-35130R1

Dear Dr. Barranco-Trabi,

We’re pleased to inform you that your manuscript has been judged scientifically suitable for publication and will be formally accepted for publication once it meets all outstanding technical requirements.

Kind regards,

Sanjay Kumar Singh Patel, Ph.D.

Academic Editor

PLOS ONE

Reviewers' comments:

Reviewer's Responses to Questions

**Comments to the Author**

1. If the authors have adequately addressed your comments raised in a previous round of review and you feel that this manuscript is now acceptable for publication, you may indicate that here to bypass the “Comments to the Author” section, enter your conflict of interest statement in the “Confidential to Editor” section, and submit your "Accept" recommendation.

Reviewer #1: All comments have been addressed

Reviewer #2: All comments have been addressed

Reviewer #1: This article entitled "PERSONS TESTED FOR SAR-CoV-2 AT A MILITARY TREATMENT FACILITY IN HAWAII" has improved.

Reviewer #2: In this paper entitled "Persons tested for SAR-CoV-2 at a military treatment facility in Hawaii." the authors aimed to identify trends in race-based disparities in COVID-19 in the setting of universal access to care. There is huge improvement in the manuscript from the past version. In addition, the author has addressed all the comments in the manuscript. The manuscript looks much better now for publication. There is no technical limitation that can be held responsible for the rejection of the manuscript. I congratulate the authors for the work.

---

## [Editor Report · Acceptance letter]

27 Jan 2022

PONE-D-21-35130R1 

Persons tested for SAR-CoV-2 at a military treatment facility in Hawaii 

Dear Dr. Barranco-Trabi:

I'm pleased to inform you that your manuscript has been deemed suitable for publication in PLOS ONE. Congratulations! Your manuscript is now with our production department. 

Kind regards, 

on behalf of

Dr. Sanjay Kumar Singh Patel 

Academic Editor

PLOS ONE